# Vision-Based Detection of Bolt Loosening Using YOLOv5

**DOI:** 10.3390/s22145184

**Published:** 2022-07-11

**Authors:** Yuhang Sun, Mengxuan Li, Ruiwen Dong, Weiyu Chen, Dong Jiang

**Affiliations:** School of Mechanical and Electronic Engineering, Nanjing Forestry University, Nanjing 210037, China; sunyh@njfu.edu.cn (Y.S.); lmx0727@njfu.edu.cn (M.L.); drw@njfu.edu.cn (R.D.); wychen@njfu.edu.cn (W.C.)

**Keywords:** bolt loosening, deep learning, machine vision, YOLOv5

## Abstract

Bolted connections have been widely applied in engineering structures, loosening will happen when bolted connections are subjected to continuous cyclic load, and a significant rotation between the nut and the bolt can be observed. Combining deep learning with machine vision, a bolt loosening detection method based on the fifth version of You Only Look Once (YOLOv5) is proposed, and the rotation of the nut is identified to detect the bolt loosening. Two different circular markers are added to the bolt and the nut separately, and then YOLOv5 is used to identify the circular markers, and the rotation angle of the nut against the bolt is calculated according to the center coordinate of each predicted box. A bolted connection structure is adopted to illustrate the effectiveness of the method. First, 200 images containing bolts and circular markers are collected to make the dataset, which is divided into a training set, verification set and test set. Second, YOLOv5 is used to train the model; the precision rate and recall rate are respectively 99.8% and 100%. Finally, the robustness of the proposed method in different shooting environments is verified by changing the shooting distance, shooting angle and light condition. When using this method to detect the bolt loosening angle, the minimum identifiable angle is 1°, and the maximum detection error is 5.91% when the camera is tilted 45°. The experimental results show that the proposed method can detect the loosening angle of the bolted connection with high accuracy; especially, the tiny angle of bolt loosening can be identified. Even under some difficult shooting conditions, the method still works. The early stage of bolt loosening can be detected by measuring the rotation angle of the nut against the bolt.

## 1. Introduction

Bolted connections have been widely used in engineering structures because of the advantages of convenient assembly and disassembly. When the structure is subjected to continuous vibration load, the bolted connection will loosen and the reliability will decrease, which may lead to structural failure [1,2,3,4]. It is of great significance to investigate the detection method of bolt loosening [5].

The mechanical performance of the bolted connection is related to pretightening torque, thread pair contaction and friction of contact surfaces [6,7,8]. In order to explore the mechanism of bolt loosening, Goodier et al. [9] tested the behavior of bolted connections under dynamic load. They explained that bolt loosening is caused by relative motion between screw threads and fasteners. Junker [10] designed classic experimental equipment to reveal that the relative motion between screw threads and fasteners is the main cause of bolt loosening. The decrease in pretightening torque and the rotation of bolts or nuts are the main phenomena of loosening. Jiang et al. [11] investigated the decline in preload in the loading cycle. The ratio relationship between the current preload and the initial preload was obtained. At the same time, Junker’s experimental equipment was improved, and a component was added to measure the relative angle between the bolt and the nut. The pretightening torque and the relative angle become important in the detection of bolt loosening. Yin [12] and Huo et al. [13] proposed a new bolt loosening detection method based on a piezo-electric transducer (PZT). The actual contact area can be determined by detecting the ultrasonic wave energy transmitted between contact surfaces. Experimental results show that there is an approximately linear relationship between the signal peak value and pretightening torque of the bolt under a saturation problem, which can be used to monitor bolt loosening. Xu et al. [14] proposed an improved time reversal method to monitor bolt loosening, which reconstructs the phase and amplitude of the signal. If there is a phase shift and amplitude difference between the signal generated by the structure in the healthy state and the reconstructed signal, it indicates that the structure is damaged. Experiments show that this method can realize quantitative monitoring of bolt preload with higher precision and sensitivity. Zhao et al. [15] combined PZT with the time reversal method for real-time health monitoring of bolted connections in wood structures. A bolt pretightening torque loss index of wood structure was proposed based on wavelet analysis design, which can reflect the looseness of bolts in wood structures. Zhang et al. [16] proposed a bolt loosening detection method based on audio classification. By recording and extracting hammer sounds of bolted connections at different loosening degrees, support vector machine (SVM) was used to train and test datasets, and quantitative detection results of bolt loosening were finally obtained. This method has high recognition accuracy and strong anti-noise ability. Wang et al. [17] proposed a new vibroacoustic method (VAM) for detecting the looseness of multi-bolt connections. The above detection methods can basically achieve unmanned online monitoring of bolted connections. However, these detection methods all need specific sensors to collect signals [18,19,20] and will undoubtedly increase the cost and difficulty of monitoring in the bolted connection structures due to the increasing number of sensors.

With the development of camera and image processing technology, structural health monitoring methods based on machine vision have been developed rapidly. Kromanis et al. [21] proposed a vision-based test method for measuring deformation and cracks of reinforced concrete structures. In addition, Kromanis also proposed a damage detection technique for bridge structure based on computer vision-derived parameters [22]. Cameras were used to collect the image frames of the bridge model under traffic loads, and the nodal displacements of the bridge model were computed from each image frame by an image processing algorithm. Structural responses such as deflection and strain were calculated according to the nodal displacements. Finally, the damage of the bridge structure was detected by analyzing the structural response. Kromanis’ research shows that the machine vision-based methods for structure health monitoring are more efficient and less costly than traditional monitoring methods. Many researchers have studied the loosening of bolt connections based on machine vision and image processing technology. Huynh et al. [23] proposed a method to identify the rotation angle of the nut using the Hough transform algorithm and to detect whether the bolt is loose by comparing the angle changes before and after. This method based on visual image can detect the nut rotation angle with an accuracy of ±2.6°. With the rapid development of deep learning, various neural networks have emerged with high recognition accuracy. For example, AlexNet [24] used GPU to accelerate computing for the first time, and other networks such as VGGNet [25], R-CNN [26], Fast R-CNN [27], GoogleNet [28] and Faster R-CNN [29] have continuously improved the recognition accuracy of target detection. In addition, YOLO [30] and SSD [31] considered the speed and accuracy of recognition. Compared with traditional methods, methods based on deep learning can autonomously learn the characteristics of data [32,33,34,35,36]. In the field of bolt loosening detection, Zhuang et al. [37] combined the time reversal method with deep learning methods to classify the ultrasonic signals in the bolted connections of wood structures, thus realizing the prediction of residual preload on bolted connections. Cha and Choi et al. [38,39,40] combined machine vision with support vector machine (SVM) to automatically distinguish tight bolts and loose bolts by detecting horizontal and vertical lengths of bolt heads in images. Huynh et al. [41] used R-CNN to detect and cut plausible bolts in bolt images, and then the Hough linear transformation (HLT) image processing algorithm was used to automatically estimate the angle of bolt loosening from bolt images. Zhao et al. [42] used SSD to identify bolt heads and the numbers on bolt heads, and the included angle of the center coordinates of the two predicted boxes was calculated. The monitoring of bolt loosening can be realized by measuring the change of the angle. The minimum identifiable angle of the method is 10°, and the angle of bolt loosening can be detected by 360°. Zhang et al. [43,44] used Faster R-CNN to train different screw heights after bolt loosening to determine whether bolts are tight or loose, and the recognition accuracy reached 95.03%. Pham et al. [45] used composite bolt images generated by graphical models as datasets trained by a neural network, which is helpful in reducing the time and cost of collecting high-quality training data. Pal et al. [46] extracted identification features using convolutional neural network (CNN) from time-frequency scale images based on vibration to detect bolt loosening. The average accuracy of the method is respectively 100% and 98.1%. Pan et al. [47] proposed an RTDT-bolt method by combining YOLOv3-tiny with optical flow method. The method achieved real-time detection and tracking of bolt rotation with an accuracy of more than 90%. Yuan et al. [48] used MASK R-CNN to complete the identification and classification of bolt loosening in near real time through a webcam. The minimum identifiable screw height was 4 mm. Gong et al. [49] proposed a bolt loosening detection method combining deep learning with geometric imaging theory, which can accurately calculate the length of exposed bolts. First, the exposed bolt was located using Faster R-CNN, and then, five key points on the exposed bolt were identified using CPN. Finally, the length of the exposed bolt was calculated by a length calculation module. The mean measurement error of this method is only 0.61 mm. The above detection methods combine deep learning with machine vision, which can not only identify various features of bolted connections but can also detect bolt loosening more intuitively and with higher precision. However, some of the above methods can only distinguish tight bolts and loose bolts, failing to determine the loosening degree of bolted connections. When the bolt loosening angle is tiny, the above methods cannot realize the early monitoring of bolt loosening. Therefore, it is necessary to investigate the detection method of bolt loosening angle.

Bolt loosening in engineering structures can be transformed into a target detection problem. Combining deep learning with machine vision, a bolt-loosening detection method based on a neural network is proposed. By adding two different circular markers on the bolted connection, a neural network is used to detect the included angles between the markers. The detection of bolt loosening can be realized by calculating the rotation angle of the nut against the bolt. Due to the small size of the markers on the bolt, YOLOv5 is effective and more efficient in detecting small targets. Compared with YOLOv3 [50,51] and v4, the network structure of YOLOv5 can extract deeper features and achieve better detection results. First, bolt images were collected using a smartphone and were trained by YOLOv5. Then, the trained model was used to detect the rotation angle of the nut against the bolt, and experiments were carried out in different environments to verify the detection accuracy of the proposed method.

## 2. Problem Description

At present, most research on the mechanics of bolt loosening is mainly based on the bolt loosening experimental device designed by Junker [3], as shown in Figure 1a. Jiang [4] obtained the experimental curve of bolt loosening through experiments, as shown in Figure 1b. P is the bolt pretightening torque, and θ is the nut rotation angle. In stage Ⅰ of the bolt preload curve, the preload decreases mainly due to material deformation, and the nut rotation angle is small. In stage Ⅱ, the preload begins to decline rapidly, while the nut rotation angle increases. The bolted connection loosens, which leads to structural failure. As shown in Figure 1b, when the preload starts to decline rapidly, the nut rotation angle is less than 5°. The nut rotation angle increases to 20° when the preload drops to 20%. Thus, an obvious rotation of the nut can be observed when the bolted connection is loosened. Therefore, combining deep learning with machine vision to detect the rotation angle of the nut against the bolt is feasible to diagnosis the loosening of bolted connections.

## 3. Methodology

### 3.1. YOLOv5 Algorithm for Bolt Loosening

The core idea of YOLO is to take the entire image as the input of the network and to directly obtain all bounding boxes of all categories through a forward propagation, so as to achieve One-Stage. First, the bolt image is divided into S×S grids, and each grid unit predicts B bounding boxes and their confidence, which roughly covers the entire bolt image. Each bounding box contains five predicted values (x, y, w, h, c), where (x, y) is the center coordinate of the bounding box, and w and h are the width and height of the bounding box relative to the entire bolt image, respectively. c stands for confidence and represents the intersection over union (IOU) of the predicted box and the ground truth. Each grid unit also predicts the probability C of the category of the object when it appears and then realizes the final detection through B bounding boxes and confidence predicted by each grid unit, as well as the class probability map. Its detection principle is shown in Figure 2.

The network structure of YOLOv5 is shown in Figure 3. The Mosaic data enhancement method is adopted in the input of YOLOv5. Images are spliced through random scaling, random clipping and random arrangement, which has a good detection effect on small targets. Adaptive anchor frame computation is added in YOLOv5. In the training of the network, the predicted boxes are output on the basis of the initial anchor frame and are compared with the ground truths. The gaps between them are calculated, and then the network parameters are reversely updated and iterated. YOLOv5 can adaptively calculate the value of the optimal anchor frame in different training sets in each training. Focus structure is added in the backbone of YOLOv5 to realize slice operation, which can fully extract features and retain more complete downsampling information of pictures. In addition, the CSP structure can increase the gradient value of backpropagation between layers and avoid the gradient disappearance caused by deepening. The structure of FPN+PAN is used in the neck of YOLOv5. Feature pyramid networks (FPN) can transfer and fuse high-level feature information through upsampling to obtain feature maps for prediction. On this basis, a path aggregation network (PAN) structure is added to improve the propagation speed of low-level features. The structure of FPN+PAN enhances the detection capability of the model for objects of different scales and can realize the recognition of the same objects of different sizes and proportions. GIOU_Loss is adopted at the output of YOLOv5 as the loss function of the bounding boxes, and the non-maximum suppression (NMS) operation is used in the post-processing of target detection to screen out the bounding boxes with the highest probability.

### 3.2. The Detection Method of Bolt Loosening

Images of the bolted connection are captured using a smartphone and are taken under different shooting distances, different shooting angles and different light conditions. In these experiments, a total of 200 images (3024 × 3024 pixels) are collected; the image format is JPG. Smartphone camera specifications are shown in Table 1. The labeling tool named ‘LabelImg’ is used to label 200 images. The bolt is defined as ‘bolt’, the blue circular marker on the nut is defined as ‘nut’, and the red circular marker on the bolt is defined as ‘sign’. Figure 4b is the graph after labeling the three classes. These labeled images are then made into the dataset.

The dataset contains three classes, namely ‘bolt’, ‘nut’ and ‘sign’. The YOLOv5 algorithm under the Pytorch framework was used to train the dataset. Then, the YOLOv5 can identify and locate the three classes and output the center coordinates of the three classes’ predicted boxes. As shown in Figure 5, the top left corner of the image is taken as the origin, the center of the ‘bolt’ class is known to be $A\left(x_{1},y_{1}\right)$, the center of the ‘nut’ class is $B\left(x_{2},y_{2}\right)$ and the center of the ‘sign’ class is $C\left(x_{3},y_{3}\right)$. Based on these coordinates, the angle can be calculated by the vector product.

The angle with A as the vertex is
(1)$$∠A=\arccos\left(\;\frac{\vec{AB}\cdot \vec{AC}}{\left|\vec{AB}\right|\cdot \left|\vec{AC}\right|}\right)$$

When a bolt is loosened, the ∠A changes with the rotation angle of the ‘nut’ class, and the degree of the bolt loosening is determined by detecting the change of ∠A. The change of ∠A is ∆. The expression of ∆ is shown in Equation (2).
(2)$$\Delta =\alpha -\alpha_{0}$$
where α is the angle calculated by the algorithm after the bolted connection has worked for a period of time; $\alpha_{0}$ is the angle when the bolted connection is tightened.

When ∆ begins to increase, it indicates that the bolted connection has begun to loosen. The bolted connection fails and the structure may become damaged when ∆>20. By detecting the rotation angle of the nut against the bolt and paying attention to the change of ∆, the loosening of the bolted connection can be effectively monitored, and anti-loosening measures can be taken in the early stage of bolt loosening.

## 4. Detection of Bolt Loosening

### 4.1. Model Training

The dataset used in this paper contains 200 images, 80% of which are used as a training and validation set and 20% as a test set. The training set is trained by YOLOv5 under the Pytorch framework. Since GPU acceleration calculation was used in training and the hardware device is Nvidia RTX 3070, the batch size of model training is set as 8. In the training process, the model loss will decrease with the increase in iterations until it is in a stable state. At this time, the detection accuracy of the model is also in a stable state and in a high position. Therefore, an appropriate number of iterations not only reduces the training time of the model, but also keeps good detection accuracy. Since there are only 160 images in the dataset used for training in this paper, in order to make the model more accurate, the total training cycles are set as 500 epochs. One epoch is equivalent to training all samples in the training set once. The larger the value is, the more accurate the model will be and the longer the training time will be. Parameters of network training are shown in Table 2.

The evaluation of model performance is mainly based on some parameters, such as precision, recall and mAP. Precision refers to the number of real positive samples in positive sample results detected based on the predicted results. Recall refers to the number of real positive samples in the test set based on real results. mAP is the average identification accuracy of all classes. mAP@0.5 indicates the mAP of positive samples detected when the threshold of IOU is 0.5. The expressions for precision and recall are shown below.
(3)$$\mathrm{precision}=\frac{TP}{TP+FP}$$
(4)$$\mathrm{recall}=\frac{TP}{TP+FN}$$
where *TP* is the number of positive samples detected as positive samples, *FP* is the number of negative samples detected as positive samples, and *FN* is the number of positive samples detected as negative samples.

The output of YOLOv5 adopts GIOU_Loss as the loss function of the bounding boxes. The smaller the value is, the more accurate the predicted box is. The expression of GIOU_Loss is shown in Equation (5). Obj_Loss is the mean value of objectiveness loss. The smaller the value is, the more accurate the target detection is. Cls_Loss indicates the average value of classification loss. A smaller value indicates more accurate classification.
(5)$$\{\begin{array}{c} \mathrm{GIOU}=IOU-\frac{A^{c}-U}{A^{c}}\\ \mathrm{GIOU}\_\mathrm{Loss}=1-\mathrm{GIOU} \end{array}$$
where IOU is intersection over union and represents the ratio of intersection and union between the predicted box and the ground truth. $A^{c}$ is the smallest tangent rectangle between the predicted box and the ground truth; U is the union of the predicted box and the ground truth.

The training results of the model are shown in Figure 6.

The training results show that the GIOU_Loss, Obj_Loss and Cls_Loss of the model are both small after 10,000 iterations. The precision rate and recall rate of the model are respectively 99.8% and 100%, and the value of mAP@0.5 is over 0.95. These data indicate that the model can accurately identify the three classes in the image. Although the sizes of the ‘nut’ class and the ‘sign’ class are small and the ‘sign’ class is inside the ‘bolt’ class, the model can still detect accurately and achieve high recognition accuracy. In addition, we tried to use Single Shot MultiBox Detector (SSD) to train the dataset, but the obtained model could only recognize the ‘bolt’ class but could not recognize the ‘nut’ class and the ‘sign’ class. This may be because the small scale of the feature map generated by SSD, which does not accurately identify the ‘nut’ class and the ‘bolt’ class of quite small size. YOLOv5, however, has adaptive anchors and multi-scale fusion to better handle objects of any size. Therefore, it is effective and feasible to use YOLOv5 to detect the loosening angle of bolted connections.

### 4.2. Identification of Bolt Loosening Angles

#### 4.2.1. Identification of Bolt Loosening at Any Angle

This section completes the detection of the ‘bolt’ class, the ‘nut’ class and the ‘sign’ class in the image by the pictures taken vertically. YOLOv5 is used to identify the three classes and output the center coordinates of their predicted boxes in the image. According to the calculation process described in the previous section, the nut rotation angle can be obtained, and the loosening degree of the bolted connection can be judged by detecting the change of nut rotation angle. The measurement method of bolt loosening angle is shown in Figure 7a. The nut is rotated 15°, 30°, 45° and 60°, and the identification results are shown in Figure 7. Because there is no particularly accurate method to measure the rotation angle of the nut against the bolt, a protractor is chosen to measure the nut rotation angle. Finally, the nut rotation angle measured by the protractor is compared and analyzed with the detection value of YOLOv5; analysis of the experimental data is shown in Table 3.

The error of bolt loosening angle is calculated by Equation (6).
(6)$$Error=\frac{\left|D-M\right|}{M}$$
where D is the detection value of the YOLOv5 algorithm, and M is the experimentally measured value.

The results show that when the bolt loosens at any angle, the average difference of the nut rotation angle is about 1.56° and the average error is about 1.34%. The difference and error of bolts at the above rotation angle are small, and the mAP is about 0.946, which indicates that the detection accuracy of bolt loosening by this method is enough.

#### 4.2.2. Identification of Bolt Loosening at Tiny Angle

In order to verify the effectiveness of the proposed method in the case of bolt loosening at a tiny angle, experiments are carried out within the range of bolt loosening at 10° in this section. The nut is rotated 10°, 8°, 5°, 2° and 1°. The identification results are shown in Figure 8, and analysis of the experimental data is shown in Table 4.

The results show that when the bolt loosens at a tiny angle, the average difference of the nut rotation angle is about 1.18°, the average error is about 1.27% and the mAP is about 0.942. When the nut is rotated only 1°, the error increases to 2.90%, and the minimum identifiable angle can be determined as 1°. The smaller the bolt loosening angle is, the larger the detection error is. No matter if the bolt loosening angle is arbitrary or tiny, the detection errors are both small, and the proposed method has high recognition accuracy for the three classes. Therefore, the proposed bolt loosening detection method using YOLOv5 can monitor bolt loosening effectively, and the anti-loosening measures can be taken in the early stage of bolt loosening.

### 4.3. Identification under Different Shooting Conditions

#### 4.3.1. Different Shooting Distances

In order to explore the influence of shooting distance on the detection results, the angle of 150° measured during the bolted connection tightening is taken as the initial state. The light condition and vertical shooting angle are kept unchanged; only the shooting distance of the camera is changed. Images are collected at distances of 5, 10, 15 and 20 cm between the bolted board and the camera. The identification results are shown in Figure 9, and the experimental data analysis of bolt loosening under different shooting distances is shown in Table 5.

The results show that when the shooting distance is 5 cm, although the mAP is high, the images taken by the camera are blurred, and the difference of nut rotation angle is 2.14°. When the shooting distance is within 10~15 cm, there is little difference in the detected value of the nut rotation angle. However, when the shooting distance is 20 cm, the recognition accuracy of the ‘nut’ class detected by the model is only 0.68. The recognition accuracy will decrease with the increase in the shooting distance. Therefore, the recommended shooting distance of images is 10~15 cm to achieve the best detection results when using the proposed bolt loosening detection method.

#### 4.3.2. Different Shooting Angles

The positions of bolted connections are widely distributed in the engineering; thus, it is impossible to ensure that bolt images collected are always taken from a vertical angle. In order to explore the influence of shooting angle on the detection results, the angle of 150° measured during the bolted connection tightening is taken as the initial state. According to the conclusion obtained in the previous section, the distance between the camera and the bolt was kept within the range of 10~15 cm. While maintaining the light conditions, only the shooting angle was changed by tilting the camera. Images are collected under four shooting angles, which are perpendicular to the camera and are tilted 10°, 30° and 45°. The identification results are shown in Figure 10, and the experimental data analysis of bolt loosening under different shooting angles is shown in Table 6.

The results show that the error increases to 5.91% when the camera is tilted 45°. The detection error of the nut rotation angle increases with the increase in shooting angle. Therefore, vertical shooting should be kept as far as possible to avoid large errors when using the proposed bolt loosening detection method.

#### 4.3.3. Different Light Conditions

In a real engineering environment, light conditions are also different. In order to explore the influence of light condition on the detection results, experiments are separately carried out under normal light, weak light, dark light and camera flash with the other shooting conditions unchanged. The identification results are shown in Figure 11, and the experimental data analysis of bolt loosening under different light conditions is shown in Table 7.

The results show that the detection error of nut rotation angle is minimum under normal light. The model has high recognition accuracy even in a weak light environment, and the error of bolt loosening angle is still small, only 2.79%. In dark light, none of the three classes could be detected. At this point, the camera flash can be turned on to increase the ambient brightness, such that the objects that cannot be recognized under dark light can still be recognized with high accuracy. However, due to the large light intensity of the flash lamp, the light reflection of the bolt surface is serious, which reduces the identification accuracy of the ‘bolt’ class.

In this study, when the shooting distance is changed, too close of a shooting distance will make the captured image blurred, while too far of a shooting distance will reduce the accuracy of target detection. Therefore, a suitable shooting distance is necessary to achieve the best detection effect with high precision. With the increase in shooting angle, the detection error of bolt loosening angle also increases. Therefore, vertical shooting should be maintained to achieve the best identification results. Meanwhile, different light conditions will also affect the detection results. The detection error is minimum under normal light, but the bolt loosening angle cannot be detected under dark light.

## 5. Summary and Conclusions

In this paper, a bolt loosening detection method based on YOLOv5 is proposed by combining deep learning with machine vision. YOLOv5 is used to train the model with high accuracy. The rotation angle of the nut against the bolt can be detected to realize the monitoring of bolt loosening. First, the effectiveness and feasibility of the proposed method are proven by the experiments of bolt loosening at any angle. Second, the applicability and accuracy of this method in detecting the tiny angle of bolt loosening are verified by experiments. Finally, the robustness of bolt loosening detection method based on deep learning and machine vision is verified through a series of experiments under different shooting conditions. In general, the proposed method can effectively and intuitively detect bolt loosening, and this has the advantages of high precision, high efficiency and low cost. At the same time, the detection of tiny angles is also effective, which provides a certain technical reference for the early detection of bolt loosening. The proposed method can transform the bolt loosening monitoring from manual inspection to automatic monitoring by fixed cameras.

Experimental investigations are undertaken under different shooting conditions, and the conclusions are as follows:The precision rate and recall rate of the model trained by the dataset are respectively 99.8% and 100%, and the mAP of the model is over 0.95. This method not only de-creases the time to collect real bolt images, but also improves the generalization ability of the network and reduces the cost of detection.The smaller the bolt loosening angle is, the larger the detection error is. The method is also accurate in detecting the tiny angle of bolt loosening. The minimum identifiable angle is 1°, and the error is only 2.90%.The detection error of nut rotation angle will increase with the increase in shooting angle, and the maximum error of bolt loosening angle is only 5.91% when the camera is tilted 45°. This shows that the method is effective and accurate even under some difficult shooting conditions.The detection results are not sensitive to the shooting distance and the light condition. When the shooting distance is within 10~15 cm and the light is sufficient, the detection accuracy is the best.

Since it is not always convenient for the camera to collect images directly against the bolts, how to reduce the influence of shooting angle and light conditions on the detection results is a crucial issue. The scope of application can be expanded after eliminating the limitations of the proposed method.

## Figures and Tables

**Figure 1 sensors-22-05184-f001:**
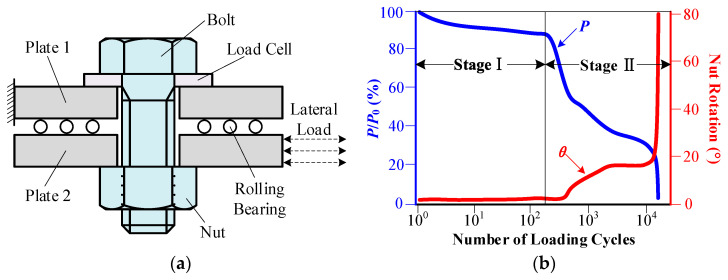
Experiment of bolt loosening: (**a**) experimental equipment of bolt loosening; (**b**) curve of bolt loosening.

**Figure 2 sensors-22-05184-f002:**
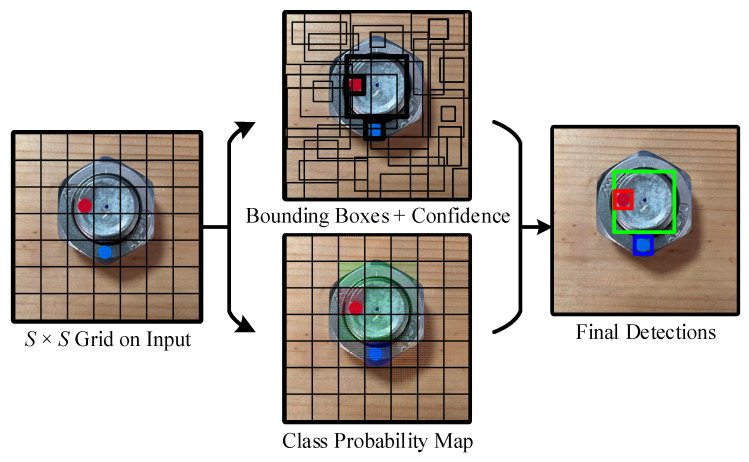
The principle of target detection using YOLO.

**Figure 3 sensors-22-05184-f003:**
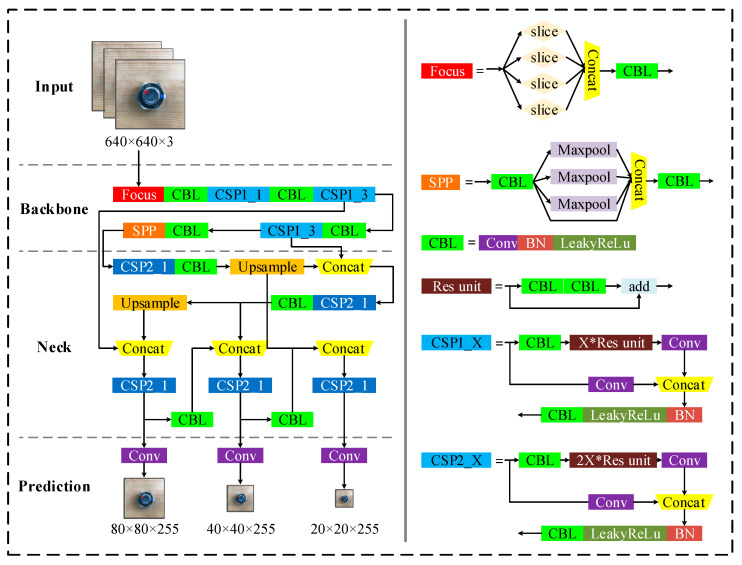
YOLOv5 network structure.

**Figure 4 sensors-22-05184-f004:**
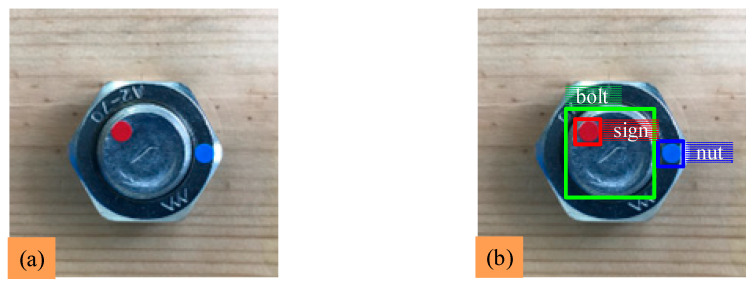
Labeling of the dataset: (**a**) captured bolt image; (**b**) labels of three classes.

**Figure 5 sensors-22-05184-f005:**
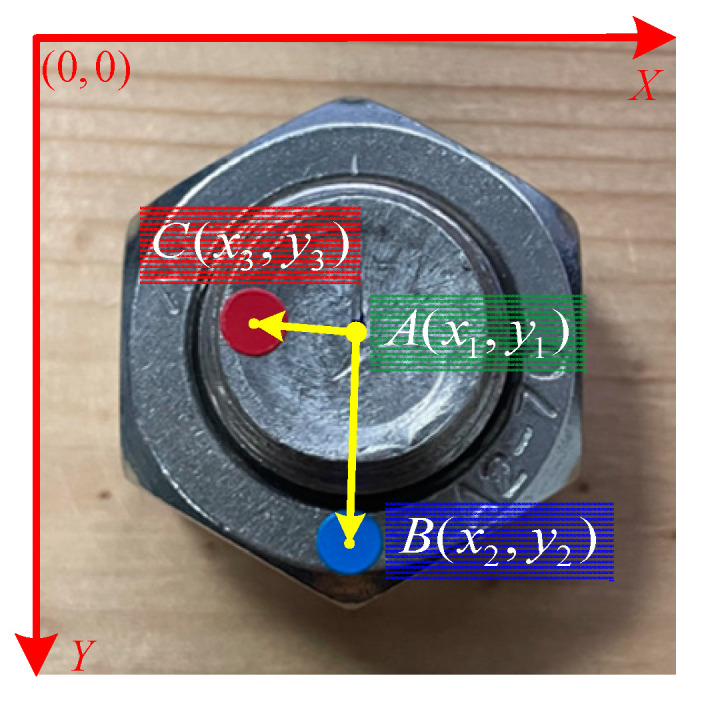
Center coordinates of the three classes.

**Figure 6 sensors-22-05184-f006:**
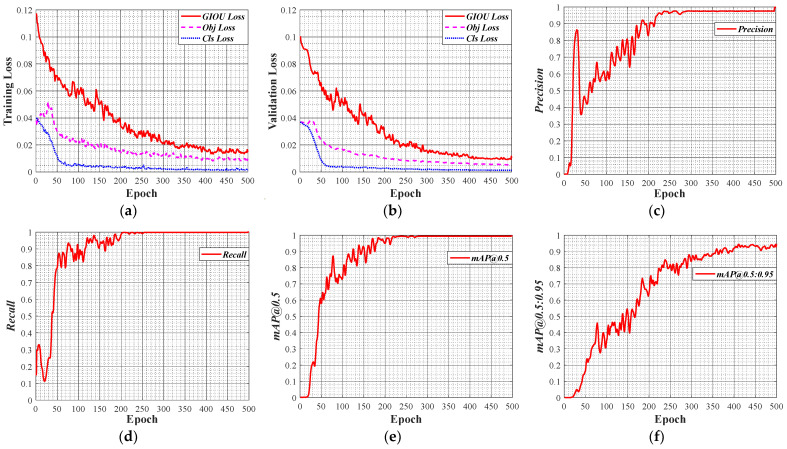
Training results of the model: (**a**) training loss; (**b**) validation loss; (**c**) precision; (**d**) recall; (**e**) mAP@0.5; (**f**) mAP@0.5:0.95.

**Figure 7 sensors-22-05184-f007:**
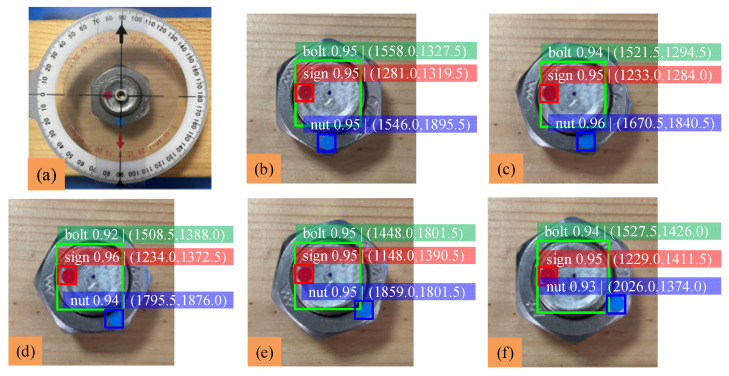
Identification results of bolt loosening at any angle: (**a**) angle-measuring method; (**b**) initial state; (**c**) 15° of rotation; (**d**) 30° of rotation; (**e**) 45° of rotation; (**f**) 60° of rotation.

**Figure 8 sensors-22-05184-f008:**
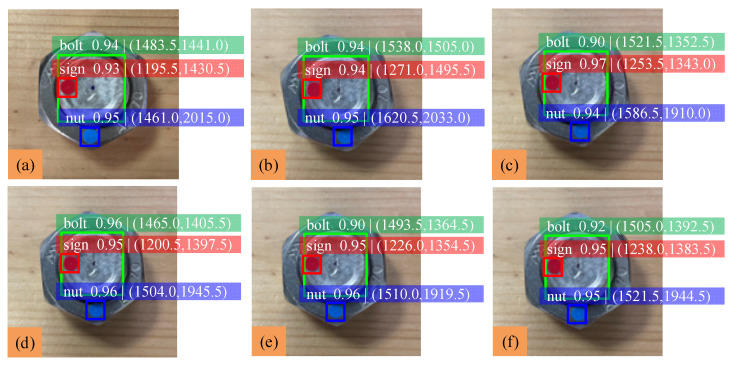
Identification results of loose bolts at tiny angle: (**a**) initial state; (**b**) 10° of rotation; (**c**) 8° of rotation; (**d**) 5° of rotation; (**e**) 2° of rotation; (**f**) 1° of rotation.

**Figure 9 sensors-22-05184-f009:**
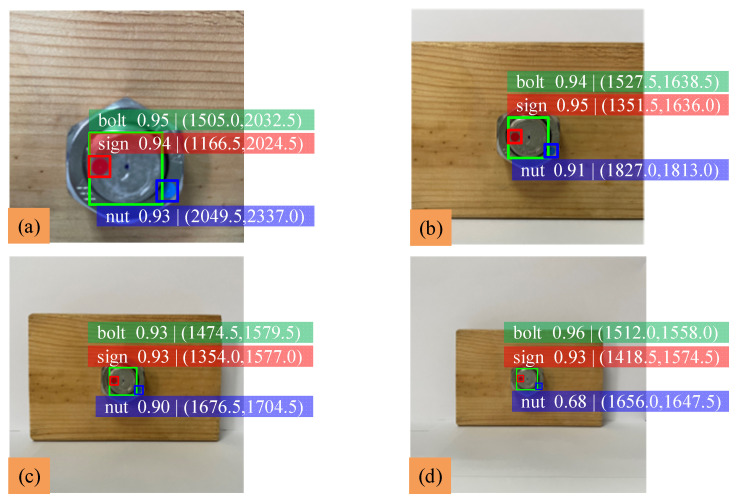
Identification results of bolt loosening under different shooting distances: (**a**) 5 cm; (**b**) 10 cm; (**c**) 15 cm; (**d**) 20 cm.

**Figure 10 sensors-22-05184-f010:**
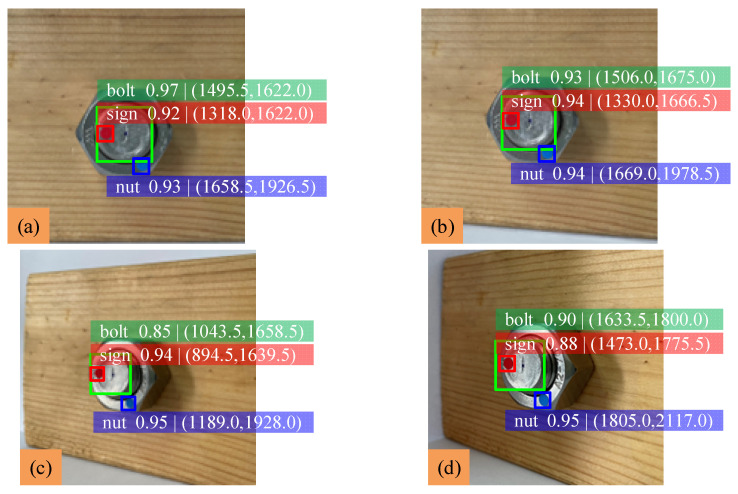
Identification results of bolt loosening under different shooting angles: (**a**) 0° of tilt; (**b**) 10° of tilt; (**c**) 30° of tilt; (**d**) 45° of tilt.

**Figure 11 sensors-22-05184-f011:**
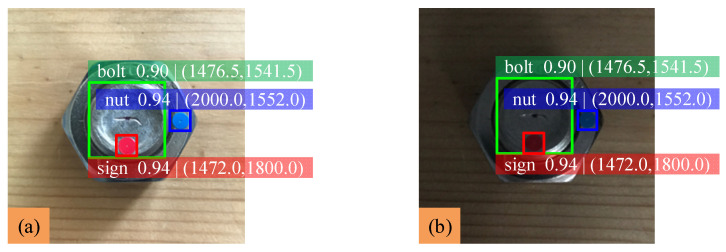
Identification results of loose bolts under different light conditions: (**a**) normal light; (**b**) weak light; (**c**) dark light; (**d**) camera flash.

**Table 1 sensors-22-05184-t001:** Smartphone camera specifications.

Parameters	Value
Size	3024 × 3024 pixels
Vertical resolution	72 dpi
Horizontal resolution	72 dpi
Bit depth	24
Aperture	f/1.8
Focal length	4 mm

**Table 2 sensors-22-05184-t002:** Parameters of training.

Parameters	Value
Image size	640 × 640
Learning rate	0.01
Momentum	0.937
Weight decay	0.0005
Batch size	8
Iteration per epoch	20
Total epoch	500

**Table 3 sensors-22-05184-t003:** Experimental data analysis of bolt loosening at any angle.

Test Sample	Rotation Angle (°)	Detection Value (°)	Measured Value (°)	Error (%)
b	0	90.44	90	0.49
c	15	107.35	105	2.24
d	30	123.69	120	3.08
e	45	135.26	135	0.19
f	60	151.07	150	0.71

**Table 4 sensors-22-05184-t004:** Experimental data analysis of bolt loosening at tiny angle.

Test Sample	Rotation Angle (°)	Detection Value (°)	Measured Value (°)	Error (%)
a	0	89.84	90	0.18
b	10	100.92	100	0.92
c	8	98.68	98	0.69
d	5	95.86	95	0.91
e	2	93.84	92	2.00
f	1	93.64	91	2.90

**Table 5 sensors-22-05184-t005:** Experimental data analysis of bolt loosening under different shooting distances.

Test Sample	Shooting Distance (cm)	Detection Value (°)	Measured Value (°)	Error (%)
a	5	152.14	150	1.43
b	10	150.59	150	0.39
c	15	149.44	150	0.37
d	20	150.28	150	0.19

**Table 6 sensors-22-05184-t006:** Experimental data analysis of bolt loosening under different shooting angles.

Test Sample	Shooting Angle (°)	Detection Value (°)	Measured Value (°)	Error (%)
a	0	118.16	120	1.53
b	10	121.00	120	0.83
c	30	125.63	120	4.69
d	45	127.09	120	5.91

**Table 7 sensors-22-05184-t007:** Experimental data analysis of bolt loosening under different light conditions.

Test Sample	Detection Value (°)	Measured Value (°)	Error (%)
a	89.85	90	0.17
b	92.51	90	2.79
c	-	90	-
d	85.57	90	4.92

## Data Availability

Not applicable.

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
