# Peer review of "Vision-Based Detection of Bolt Loosening Using YOLOv5"

_sensors, 2022, doi:10.3390/s22145184_

Round 1
Reviewer 1 Report
The paper is interesting and can be accepted after the following minor changes:
1. The English needs improvement or polished by a native speaker.
2. Figures quality can be much better or clear. In same cases the large text covered the real image.
3. Literature survey is poor and can have lot more development in the area of vision based SHM. For example Kromanis published so many in this area but hardly any of them is discussed.
4. Conclusions are too general and must be specific.
Reviewer 2 Report
The authors of submission sensors-1773551 present an interesting work on the detection of bolt loosening by computer vision. A dataset that they used to train the predictive model was images of bolts subjected to cycling loads and contained 200 images. The authors used a machine learning algorithm called YOLOv5, which is a group of compound-scaled object detection machine learning models trained on a well-recognized COCO dataset and is a part of the PyTorch library. The authors claim that they achieved very high accuracy, with a precision rate of 99.8% and a recall rate of 100%. They also perform some simple analyses of different shooting conditions. They performed a literature review, which is comprehensive enough to give the size of the study and up to date. The problem description is presented very briefly. When using the algorithm of other researchers, such as YOLO, you should clearly indicate what your unique contribution is. A proper summary section is desirable. The results of the analysis are described in detail. The method has the potential to be used in engineering practice. Overall, the paper is well-structured, legible and coherent. Several points need to be clarified. I would recommend a minor revision. Presenting such a sophisticated method requires a closer look at the available solutions and at least a small comparative analysis with other approaches.
General comments:
§ The abstract should be rewritten. Please pay special attention to your unique contribution and details that are essential for the comprehension of your method;
§ It would be best if you focused on emphasising what your unique contribution is, it is not clear;
§ The paper contains some stylistic and structural errors. You should try to avoid wordiness and limit the use of passive voice to improve readability. I would recommend proofreading;
§ Your paper lacks a proper summary. Consider adding this summary to chapter five. You can present it as "Summary and Conclusions";
§ You should extend the "Conclusions" part with a focus on your unique contribution. It is an essential part of the manuscript. The conclusions are too short in comparison with the size of the paper;
§ You should deliberate more on Figure 5.
Additional questions:
· Could you describe the boundary conditions of your method?
· Could you deliberate more on the resilience of your method, what if picture blur is significant? Such as in videos captured by drones.
Reviewer 3 Report
The authors describe a vision-based detection of the bolt loosening method by using YOLO (You Only Look Once). The research subject is exciting and industrial-oriented, but before the manuscript becomes ready for publication, the authors need to address some issues:
The authors used YOLOv5 and did not explain the difference between the algorithm of V5 and YOLO-V3 or V4, which are extensively used for the same application.
Many abbreviations (acronyms) were used in the text, but the authors did not explain them, such as IOU, CNN, FPN etc. Adding a nomenclature section or considering all abbreviations to be explained is essential.
The authors discussed the identification of bolt loosening at any angle. How can they identify the loosening if the label in the bolt or nut moved 360 degrees relative to each other?
Please explain how the error is measured? It is necessary to mention error calculation in methods and experimental details.
Bolt and nut are usually used based on the sizing class. How accurate is the method for different bolt and nut classes based on the size?
It is essential to discuss the target market of this detection method. Here is some paper as references that could be used to explain the possible market:
Yang, Xinyue, Yuqing Gao, Cheng Fang, Yue Zheng, and Wei Wang. "Deep learning‐based bolt loosening detection for wind turbine towers." Structural Control and Health Monitoring 29, no. 6 (2022): e2943.
Zhou, Jing, and Linsheng Huo. "Computer Vision-Based Detection for Delayed Fracture of Bolts in Steel Bridges." Journal of Sensors 2021 (2021).
Mehmanparast, Ali, Saeid Lotfian, and Sukumara Pillai Vipin. "A review of challenges and opportunities associated with bolted flange connections in the offshore wind industry." Metals 10, no. 6 (2020): 732.
Jiang, Anyao, and Jun Liu. "Pipeline Flange Defect Detection based on Deep Learning." In Proceedings of the 2020 International Conference on Aviation Safety and Information Technology, pp. 296-300. 2020.
The method works based on the measuring of the colour labels. Measuring the preload during the experimental campaign could prove the accuracy and efficiency of the process. Do the authors have such data or measured preload during the experiment?
Round 2
Reviewer 3 Report
The manuscript is at an acceptable level to be published, considering the revised version.